# Platelet Proteome Reveals Novel Targets for Hypercoagulation in Pseudoexfoliation Syndrome

**DOI:** 10.3390/ijms25031403

**Published:** 2024-01-24

**Authors:** Elif Ugurel, Ghazal Narimanfar, Neslihan Cilek, Cem Kesim, Cigdem Altan, Afsun Sahin, Ozlem Yalcin

**Affiliations:** 1Research Center for Translational Medicine (KUTTAM), Koc University, Istanbul 34450, Turkey; eugurel@ku.edu.tr (E.U.); ghazalnarimanfar@gmail.com (G.N.); ncilek18@ku.edu.tr (N.C.); 2Department of Physiology, School of Medicine, Koc University, Istanbul 34450, Turkey; 3Department of Ophthalmology, Koc University Medical School, Istanbul 34010, Turkey; cmkesim@gmail.com (C.K.); afsahin@ku.edu.tr (A.S.); 4Beyoglu Eye Training and Research Hospital, University of Health Sciences, Istanbul 34421, Turkey; acigdemaltan@gmail.com

**Keywords:** pseudoexfoliation syndrome, retinal vein occlusion, thrombosis, coagulation, platelet, proteomics

## Abstract

Pseudoexfoliation syndrome (PEX) is characterized by the accumulation of abnormal extracellular matrix material in ocular and non-ocular tissues, including blood vessel walls. Clot-forming dysfunction might be responsible for venous thrombosis in PEX. We investigated global coagulation, the proteome, and functions of platelets in PEX patients and aimed to determine prognostic biomarkers for thrombosis risk in PEX. Peripheral blood was collected from PEX and retinal vein occlusion (RVO) patients, and age–sex matched controls. Viscoelastic hemostasis was evaluated by rotational thromboelastometry (ROTEM). Platelet markers (CD41, CD42, CD61, and CD62p) and endothelial markers (P-selectin, E-selectin, and von Willebrand factor) were investigated by flow cytometry and ELISA, respectively. The platelet proteome was analyzed by 2D fluorescence difference gel electrophoresis followed by mass spectrometry. Clot formation time (CFT) is significantly reduced in PEX patients compared to the controls (*p* < 0.05). P-selectin levels were higher in PEX patients than in controls (*p* < 0.05); E-selectin and von Willebrand factor remained unchanged. The monitorization of CFT by ROTEM, and soluble P-selectin, may help assess thrombotic risk in PEX patients. Proteomic analysis revealed differential expression of Profilin-1 in platelets. Profilin-1 regulates the stability of actin-cytoskeleton and may contribute to impaired platelet hemostatic functions. Increased P-selectin levels together with impaired coagulation dynamics might be responsible for the thrombotic events in PEX disease.

## 1. Introduction

Pseudoexfoliation syndrome (PEX) is an age-related systemic disease characterized by the accumulation of flake-like granular material in the pupillary margin, ciliary body, lens zonules, and anterior vitreous [1,2]. In general, PEX syndrome is defined as an extracellular matrix disease that causes the progressive accumulation of abnormal fibrillary material in intraocular and extraocular tissues, including the trabecular meshwork. It is the most commonly identified cause of open-angle glaucoma and irreversible vision loss worldwide [3,4]. PEX syndrome is a stress-induced elastic microfibrillopathy caused by the overproduction and abnormal accumulation of elastic fiber components in both intra- and extraocular sites. The elevation of TGF-β levels, enhanced oxidative stress, and a change in the local balance between matrix metalloproteinases (MMPs) and tissue inhibitors of metalloproteinases (TIMPs) appear to be involved in the disorder of the fibrotic matrix with the accumulation of extracellular material [5,6,7]. A mutation in the lysyl oxidase-like-1 (LOXL-1) gene from the lysyl oxidase enzyme family, which acts as a catalyst in the cross-linking of collagen and elastin, is a strong risk factor for PEX syndrome [8,9,10].

PEX is not only an ocular syndrome but also a systemic disease characterized by abnormal production and deposition of extracellular matrix material in tissues such as the walls of the heart and blood vessels [1,11]. PEX syndrome is an independent risk factor for many cardiovascular diseases [12]. Deposition of PEX material within the vasculature with subsequent increases in vascular resistance and decreases in blood flow, vascular dysregulation, and altered parasympathetic vascular control may be implicated in the pathogenesis of cardiovascular disorders in PEX subjects [13,14,15]. PEX syndrome increases the risk for atherothrombotic vascular events, cerebrovascular disease, and venous thrombosis [11,16]. PEX has also been proposed as a potential risk factor for central retinal vein occlusion (RVO) [17,18]. Ritch et al. reported that PEX material was seen in 61% of RVO cases and suggested that RVO was strongly associated with PEX [17]. Since PEX patients with RVO do not have significant exfoliative vasculopathy in or behind the lamina cribrosa, it is highly likely that platelet dysfunction and/or disturbances in clot formation and/or fibrinolysis cause thrombotic events [19]. Therefore, platelet functions and hemostasis should be considered together in these cases. Clot formation requires an accurate fibrin polymerization process involving the proper functions of elastic microfibrils. Since elastic microfibrillopathy is seen in PEX syndrome, it has been suggested for a long time that there may be pathology in clot formation. We previously suggested that clot-forming dysfunction plays a role in the etiopathogenesis of venous thrombosis in PEX disease based on our studies on rotational thromboelastometry (ROTEM), which provides a graphical evaluation of the entire coagulation process in whole blood [20]. Important parameters of ROTEM assays include clotting time (CT), clot formation time (CFT), and maximum clot firmness (MCF) [21]. Other parameters include the amplitude of the ROTEM graphic at the 10th and the 20th minute of the coagulation, which depicts the clot firmness, and the alpha angle of the ROTEM graphic, which shows the rate of coagulation (Appendix A). In our previous study, we demonstrated that PEX patients have shorter clotting time (CT) and increased maximum clot firmness (MCF) than healthy controls, which indicates a tendency for hypercoagulation in PEX cases.

Although several studies have investigated vascular disorders and thrombotic events in PEX cases, the etiopathogenesis of the disease has not yet been fully elucidated. Considering the potential for hypercoagulability in PEX syndrome, platelet functions are likely to contribute to thrombotic events. Therefore, it is important to investigate the platelet proteome of PEX patients, which would provide insight into the hypercoagulable nature of PEX syndrome. However, all proteomic studies in PEX patients to date have been performed in the lens capsule, intraocular fluid, or trabecular meshwork cells; the platelet proteome has not been examined [8,22,23,24,25]. In the present study, we investigated the whole platelet proteome and platelet functions of the patients to determine their contribution to the etiopathogenesis of the disease. Platelet cytoskeleton plays a major role in platelet function as it maintains the discoid shape of the platelets and rapidly rearranges in response to activation. The actin dynamics that underlie these shape changes depend on a large number of actin-binding proteins, such as Profilin-1 [26]. This small actin-binding protein (15 kD) acts as an ATP exchange factor for actin and controls the insertion of new monomers to growing actin filaments. Profilin-1 is essential for integrin activation, clot retraction, and platelet function [27].

In the present study, we hypothesized that the cardiovascular risk, thrombotic events, and hypercoagulable propensity of PEX patients would be provoked by the proteomic alterations in platelets that may lead to impairments in platelet functions. We studied the platelet functions in the coagulation processes by ROTEM tests, from the formation of the clot until the end of the fibrinolysis. We applied the Extem test to evaluate the extrinsic pathway components (factors II, VII, IX, and X), Intem test to estimate the intrinsic pathway factors (factors I, II, VII to XII, and von Willebrand), and Fibtem test to assess clot formation depending on the polymerization of fibrin. We demonstrated that proteomic alterations in platelets may lead to conformational changes in the actin-based cytoskeleton associated with the activation of platelets that finally affect platelet functions in PEX patients with and without retinal vein occlusion compared with age- and sex-matched controls.

## 2. Results

### 2.1. Baseline Characteristics of Study Participants

The platelet count and the mean platelet volume (MPV) of all participants are within the normal ranges. The neutrophil/lymphocyte ratio (NLR) of the patients and controls is not pathologically high (>3) but falls within the grey zone (2.3–3.0), where it may serve as a warning of infection, inflammation, and stress. However, the mean NLR of the study groups is not statistically different. The visual acuity (log MAR) of the RVO group is significantly higher than that of the PEX group and the healthy controls (*p* < 0.001). The mean intraocular pressure (mmHg) of the study groups is not significantly different.

### 2.2. Coagulation Assessment by ROTEM

The ROTEM tests measured the entire coagulation process in whole blood samples. The Extem test showed that the clot formation time (CFT) significantly decreased in PEX and RVO patients compared with the controls (*p* < 0.05) (Figure 1B). The alpha angle (the rate of the coagulation), A20 parameter (clot firmness at the 20th minute of the coagulation), and MCF (maximum clot firmness) significantly increased in RVO patients compared with the controls (*p* < 0.05) (Figure 1C,E,F).

Intem assay revealed that the alpha angle dramatically decreased in RVO patients compared with PEX patients and the controls (*p* < 0.0001) (Figure 2C). Intem parameters of PEX patients did not significantly change. No significant result was recorded in the Fibtem test for all study groups (Figure 3). The raw data of the Extem, Intem, and Fibtem tests of all studied groups are given in Appendix A. ROTEM tracings of representative subjects from each group are shown in Appendix A.

### 2.3. The Expression Levels of Platelet Markers

The surface expression of platelet markers (CD41a, CD42b, CD61, and CD62P) was calculated according to the standard curve from the BD Quantibrit PE Phycoerythrin Fluorescence Quantitation Kit. The expression of CD41a, CD42b, CD61, and CD62p on the platelet surface was estimated by the geo-mean values corresponding to the relative amounts of monoclonal antibodies bound per platelet. The relative expression level of each platelet marker was not significantly changed between PEX patients, RVO patients, and the controls (Figure 4).

### 2.4. Proteomic Analysis

A total of 8028 spots were identified in 2D gel analysis. Gel images are provided in Appendix A. Differentially expressed spots (*n* = 16) with >2-fold change were selected among study groups (controls, PEX, and RVO patients) and analyzed by MALDI-TOF/TOF. Among them, 13 protein spots were identified, which are listed in Table 1 with their respective masses and Mascot scores. A cut-off score of 31 was calculated for the 5% confidence threshold. All proteins were examined according to their protein class and biological processes (PANTHER). Identified proteins belong to the chaperon, chromatin binding, cytoskeletal, protein modifying enzyme, protein binding modulator, and structural protein classes. Most of the proteins are involved in biological regulation and cellular processes. STRING analysis showed the functional association of the identified proteins, in which a direct interaction is known between Profilin-1 (PFN1) and actin (Figure 5). Accordingly, we selected Profilin-1 for validation experiments, which might be involved in the cytoskeletal rearrangements. Expression levels of Profilin-1 were evaluated in PEX and RVO patients, and control individuals, by Western blot analysis. Profilin-1 expression is reduced in PEX patients and significantly decreased in RVO patients compared with the controls (Figure 6, *p* < 0.001). In addition, we investigated any correlations between Profilin-1 expression levels and significantly different ROTEM parameters (Extem CFT, Extem alpha, Extem A20, Extem MCF, and Intem alpha). However, we did not find a strong correlation between Profilin-1 protein expression and ROTEM parameters. 

### 2.5. Enzyme-Linked Immunosorbent Assay (ELISA)

Soluble activation markers, P-selectin, E-selectin, and von Willebrand Factor [28], were investigated in the plasma samples of all participants. P-selectin levels were significantly elevated in PEX patients, according to the controls (Figure 7, *p* < 0.05). E-selectin and vWF levels were not significantly changed among the study groups. However, a notable variation is observed in the vWF levels of PEX patients (Figure 7). In addition, we investigated any correlation between P-selectin levels and significantly different ROTEM parameters (Extem CFT, Extem alpha, Extem A20, Extem MCF, and Intem alpha). Accordingly, Extem A20 and Extem MCF are positively correlated with sP-selectin levels in the PEX group, *p* = 0.037 (r = 0.377) and *p* = 0.036 (r = 0.39), respectively.

## 3. Discussion

In the present study, we demonstrated platelet contributions to the hypercoagulable nature of PEX syndrome based on the findings that clot formation time is reduced and clot firmness is increased in PEX patients. The surface expression of platelet markers is not significantly different from that in the controls; however, soluble P-selectin levels showing in vivo platelet activation are higher in PEX patients than in controls. Proteomic analyses by mass spectrometry reveal that Profilin-1 is differentially expressed in platelets of the patients. Bioinformatic analyses show that Profilin-1 interacts directly with beta-actin, an essential element of the cytoskeleton. We demonstrated that Profilin-1 is significantly reduced in RVO patients compared to the controls.

Since PEX disease is associated with the risk of systemic vascular disease, an evaluation of inflammation, platelet count, and platelet characteristics in PEX patients is recommended for the determination of thrombotic risk. For instance, mean platelet volume (MPV) determines platelet production and activation rate, and may be responsible for abnormal platelet functions in PEX syndrome. Studies showed that MPV and the neutrophil/lymphocyte ratio (NLR) are elevated in PEX patients, and the latter is significant in the early onset of the disease [29,30]. The present study recorded no significant changes in platelet count, MPV, or NLR among the study groups; however, NLR is slightly increased in PEX patients. We further investigated the surface expression of platelet markers (CD41, CD42, CD61, and CD62p) to evaluate the hemostatic functions of platelets for their activation and adhesion on the endothelium. CD41, CD42, CD61, and CD62p are a group of glycoproteins in the platelet membrane, which refer to GPIIb, GPIb, GPIIIa, and P-selectin, respectively. Upon activation, platelets express a large amount of P-selectin, which is rapidly released from α-granules to the platelet surface [31,32]. CD41 and CD61 are platelet-specific glycoproteins that form the GPIIb/GPIIIa complex, a receptor for fibrinogen and other adhesive proteins [33,34]. We did not observe significant differences in the surface expression of platelet activation markers between PEX and RVO patients or control individuals. However, plasma levels of soluble P-selectin (sP-selectin) were significantly higher in PEX patients than in the controls. sP-selectin has been proposed as a more reliable marker of in vivo platelet activation than P-selectin on the platelet surface by flow cytometry [35]. Following platelet activation, sP-selectin is expressed on the surface membrane and then shed by cleavage [36,37]. An increased level of sP-selectin is a major predictive factor of cardiovascular and thrombotic events relating to platelet turnover and its activation and function [38,39]. In addition, sP-selectin levels are positively correlated with Extem A20 and Extem MCF in the PEX group. Higher sP-selectin levels in PEX patients increase the maximum clot firmness and the amplitude of the coagulation at the 20th minute. Neither the present study nor previous studies showed significant differences in E-selectin and von Willebrand factor levels between PEX patients and the controls [28,40]. E-selectin and von Willebrand factor are biomarkers of endothelial damage or dysfunction that can be observed in PEX pathology [41,42]. According to the results of our study, we suggest that redundant activation of platelets might contribute more to the etiopathogenesis of PEX syndrome than endothelial factors. sP-selectin in the plasma of PEX patients might represent a valid tool to assess in vivo platelet behavior in terms of predicting the risk for thrombosis.

In the present study, rotational thromboelastometry (ROTEM) analysis was performed to assess viscoelastic hemostasis in whole blood and determine the coagulation dynamics in PEX patients. Clot formation time (CFT) in Extem is significantly decreased in PEX patients compared with the controls. The reduction in CFT indicates that PEX patients have the potential to form clots in a shorter time. In addition, the alpha parameter in the Intem test was not notably changed between PEX patients and the control group but substantially decreased in the RVO group. The alpha parameter gives information about the kinetics of clot formation. For instance, a relatively large alpha angle indicates rapid coagulation [43]. One can conclude that the coagulation process by the intrinsic pathway is slower in RVO patients than in PEX patients and the controls. According to our results, we need to emphasize the propensity for PEX patients to hypercoagulation. In our previous study, we demonstrated that the clotting time (CT) in the Extem test was lower in PEX patients than in the controls [20]. The study shows that the clotting process in the beginning of the coagulation is faster in PEX cases that may be responsible for thrombotic events occurring in the patients. In another study, we demonstrated that CT and CFT (clot formation time) in RVO cases were significantly decreased compared with those in the controls [44]. The shortening of clotting time and clot formation time indicates a tendency toward hypercoagulation in PEX and RVO patients. In the present study, we did not record any significant change in CT paramater as in the previous studies. However, the significant reduction in CFT parameter in the PEX group indicates that clot is rapidly formed in the patients. The present study with a large sample size has more potential than our previous studies for tracing ROTEM parameters. Therefore, we suggest that the CFT parameter in the Extem test can be evaluated in terms of monitoring the hypercoagulable state in PEX syndrome, which may assess the risk for thrombotic events in these patients.

Proteomic and bioinformatic analyses reveal that differentially expressed proteins in platelets of PEX patients mostly belong to the cytoskeletal, protein modifying enzyme, protein binding modulator, and structural protein classes. According to the list of differentially expressed proteins, Profilin-1, iron–sulfur cluster assembly enzyme, carbohydrate deacetylase, CUSTOS, Calponin-3, and 5’-AMP-activated protein kinase subunit beta-1 have high score values (>31) by mass spectrometry analysis. Among these proteins, Profilin-1, which is likely to play a role in platelet functions and might be related to the pathology of PEX disease, was selected for validation studies by Western blot. We demonstrated that Profilin-1 is reduced in PEX patients and significantly decreased in RVO patients compared with the controls. Profilin-1 directly interacts with beta-actin, which is involved in the cytoskeletal structure and plays a key role in the shape change of platelets during activation [45,46]. Reorganization of the cytoskeleton during platelet activation enables the platelets to transition from discoid to spherical shapes. Subsequently, the cells take on an irregular shape due to the formation of pseudopods that allow platelets to spread and adhere to the damaged endothelium [47]. Actin polymerization is responsible for this shape change in platelets. Profilin-1 binds to globular actin monomers (G-actin) and separates them from filamentous actin monomers (F-actin), leading to their enhancement [48,49]. The most important function of Profilin is to renew the ATP-actin pool in platelets by providing ADP-ATP nucleotide exchange in G-actin [50]. The Profilin-1-ATP-actin complex interacts with the rapidly elongating actin filament and leads to the release of the ATP-actin monomer, allowing actin to participate in the filament structure [51]. Sritt et al. demonstrated that Profilin-1 knock-out platelets showed severely delayed and less effective clot retraction, reflecting impaired integrin function and actin dynamics [27]. Their results revealed that defective organization of the cytoskeleton in Profilin-1 knock-out platelets leads to accelerated integrin inactivation and, hence, impaired platelet function. Reduced expression of Profilin-1 in PEX and RVO patients in the present study suggests that platelet hemostatic function mediated by actin polymerization may be impaired in these patients. However, it is unclear whether impaired actin polymerization due to reduced Profilin-1 can lead to a prothrombotic phenotype. In addition, impaired coagulation dynamics in patients may occur through different mechanisms regardless of Profilin-1 function. We found no correlation between Profilin-1 expression and ROTEM parameters, suggesting that Profilin-1 has no direct effect on clot formation time, clot firmness, or coagulation rate. Elevated sP-selectin levels in PEX patients showing platelet activation might contribute more to the prothrombotic phenotype seen in the patients.

## 4. Materials and Methods

### 4.1. Study Approval

Participants of this study attended the Department of Ophthalmology, Koç University Hospital, Istanbul, Turkey, and the Beyoğlu Eye Training and Research Hospital, Istanbul, Turkey. Informed consent was provided for all participants prior to sample collection. The studies were approved by the local medical ethics committee (IRB1.013/2017) and performed in accordance with the Code of Ethics of the World Medical Association (Declaration of Helsinki) for experiments involving humans.

### 4.2. Characteristics and Demographics of the Subjects

All samples of patients and controls were collected from September 2019 until March 2021. The main characteristics of all participants are demonstrated in Table 2. All study subjects underwent a comprehensive ophthalmic examination prior to the blood sample’s collection. The patients who have PEX material on the anterior side of the lens are included as the PEX patient group (*n* = 29). The patients who were diagnosed with retinal vein occlusion (RVO) are included as the RVO patient group (*n* = 11). Age–sex matched individuals, who applied to the clinic and were not diagnosed with an ocular disease, were included as the control group (*n* = 42). Sample size was determined at 90% power according to the results of our previous study (20). Subjects with liver disease, von Willebrand disease, thrombocyte dysfunction, diabetes mellitus, intraocular pressure abnormalities, or patients who used anticoagulant and antiplatelet drugs were excluded.

### 4.3. Blood Sampling

Peripheral blood was collected from the forearms of all subjects into K3EDTA or sodium citrated vacutainers, where appropriate. Apyrase was added to the blood samples (0.02 U/mL) for the flow cytometry experiments and studied on the same day. Blood samples for ROTEM tests were studied within two hours after blood collection. Plasma samples were collected by centrifugation and stored at −80 °C for further analysis.

### 4.4. Rotational Thromboelastometry (ROTEM)

A rotational thromboelastometry device (ROTEM, Pentapharm Gmbh, Munich, Germany) was used to analyze the whole blood viscoelastic hemostasis to assess global coagulation from the beginning of clot formation to the end of fibrinolysis. Viscoelastic coagulation tests not only evaluate the contribution of plasma proteins in hypercoagulability states but also cellular contributions. ROTEM tests include Extem, Intem, and Fibtem tests, which examine the extrinsic pathway, intrinsic pathway, and fibrinolysis, respectively. All ROTEM tests were performed according to the manufacturer’s recommendations. Briefly, blood in sodium citrated tubes and test reagents were heated up to 37 °C before the study. Pins were attached to the measuring station of the device, and the cups were placed in the cup holders. The coagulation was initiated in the cups by mixing the blood with the reagents specific for each ROTEM test, and the cup holders were immediately attached to the pins. ROTEM parameters (clotting time (CT), clot formation time (CFT), alpha angle, maximum clot firmness (MCF), amplitude at 10 min (A10), and amplitude at 20 min (A20)) were analyzed for each ROTEM test.

### 4.5. Platelet Separation and Flow Cytometry

Platelets were separated from platelet-rich plasma (PRP), as previously described [52]. Briefly, citrated blood was centrifuged at 150 g for 15 min, and the upper 2/3 of the plasma was pipetted into a new tube. Apyrase (0.02 U/mL) was added to prevent the spontaneous activation of platelets. PRP was achieved after a second centrifugation step and further centrifuged at 1500 g for 10 min. After the supernatant was discarded, platelets were washed twice with citrate buffer solution (11 mM glucose, 128 mM NaCl, 4.3 mM NaH2PO4, 7.5 mM Na2HPO4, 4.8 mM sodium citrate, 2.4 mM citric acid, 0.35% (*w*/*v*) bovine serum albumin, and pH 6.5) and suspended in 500 µL FACSFlow solution (Becton Dickinson, Franklin Lakes, NJ, USA). Quantibrit PE Phycoerythrin Fluorescence Quantitation Kit (BD Biosciences, Franklin Lakes, NJ, USA) was used to establish a standard curve for the exact measurement of the level of antibodies bound per cell. Antibodies were chosen to selectively bind to the markers on the platelet surface to facilitate the analysis of the expressed receptors according to previous protocol [53]. Accordingly, PE-labeled anti-human CD41a (BD Pharmingen, Cat. No. 555467, Franklin Lakes, NJ, USA), PE-labeled anti-human CD42b (BD Pharmingen, Cat. No. 555473, Franklin Lakes, NJ, USA), PE-labeled anti-human CD61 (BD Pharmingen, Cat. No. 555754, Franklin Lakes, NJ, USA), PE-labeled anti-human CD62P (BD Pharmingen, Cat. No. 550561, Franklin Lakes, NJ, USA), and PE-labeled anti-human anti-PAC-1 antibody (Santa Cruz Biotech, Cat. No. sc-32776, Heidelberg, Germany) were studied in a 1:5 dilution. The platelet suspensions were incubated with each antibody for 20 min at room temperature in the dark. The platelets were washed once with FACSFlow solution and analyzed in PE channel by Accuri^®^ C6 flow cytometer (Becton Dickinson, Franklin Lakes, NJ, USA). In addition, FITC-labeled anti-human CD45 (BD Pharmingen, Cat. No. 555482,Franklin Lakes, NJ, USA) was used to show the lack of leukocytes in the isolated sample with a high percentage of platelets.

### 4.6. Enzyme-Linked Immunosorbent Assay (ELISA)

Endothelial activation markers in plasma were investigated by Human sP-selectin ELISA Kit (Invitrogen, Waltham, MA, USA), Human sE-selectin ELISA Kit (Invitrogen, Waltham, MA, USA), and Human vWF ELISA Kit (Invitrogen, Waltham, MA, USA) according to the manufacturer’s recommendations. Briefly, 100 μL from each standard and 20 μL from plasma samples were added into specified wells on 96-well plates coated with human P-selectin, E-selectin, and von Willebrand antigens. Then, 50 μL HRP-conjugate was mixed with the samples and standards, and incubated for 2 h. TMB substrate solution was added to each well and incubated for 10 min in the dark. The reaction was terminated by adding 100 μL stop solution. All samples and standards were studied in duplicate, and the plates were measured at 450 nm by a spectrophotometer (BioTek, Winooski, VT, USA). The concentration of each activation marker was calculated according to the formula of the absorbance–standard curve.

### 4.7. Protein Extraction from Platelets

Protein extraction from platelets was performed by the addition of the lysis buffer (7 M urea, 2 M thiourea, and 4% CHAPS) and incubated on ice for 1 h. The sample was vortexed for 30 s every 15 min and finally centrifuged at 20,000× *g* for 20 min. The supernatant was collected in a clean tube and used for subsequent analysis. First, the concentration of protein samples was determined according to Bradford method using Bradford 1× Quick Dye Reagent (BioRad, Hercules, CA, USA). An equal amount of protein from each sample of the same group was mixed for the preparation of pooled samples. Next, protein samples in each pool were precipitated by ReadyPrep 2-D Cleanup Kit (BioRad, Hercules, CA, USA), and protein concentration was determined again according to Bradford method by Bradford 1× Quick Dye Reagent (BioRad, Hercules, CA, USA). Protease inhibitors (cOmplete, Roche, Switzerland) and phosphatase inhibitors (PhosSTOP, Roche, Switzerland) were added into protein samples according to the manufacturer.

### 4.8. 2-D Fluorescence Difference Gel Electrophoresis (2DIGE)

An equal amount of protein samples was mixed for the preparation of the protein pool of each study group. Pooled protein samples were labeled with minimal CyDyes (Cy2, Cy3, or Cy5) according to the instructions provided by the manufacturer (Amersham, Cytiva Life Sciences, UK). Briefly, 50 μg protein from each pool was labeled with 400 pmol CyDye (Cy3 or Cy5) and incubated in the dark for 30 min. Randomization of samples across gels was performed by labeling each group with a different CyDye to avoid systematic errors such as experimental conditions, sample handling, and labeling. In addition, an equal amount of protein from each pool was mixed for the preparation of an internal standard, which was labeled with Cy2 in the same conditions. The internal reference was included in each assay. The labeling reaction was terminated by adding 1 μL of 10 mM lysine. The 2× DIGE sample buffer (7 M urea, 2 M thiourea, 4% CHAPS, 2% Bio-Lyte 3/10 ampholyte, and 130 mM DTT) was added to the same volume as the sample, and the pooled samples were applied to 7 cm pH 3-10 IPG strips (BioRad, Hercules, CA, USA) by active rehydration at 50 V overnight in a Protean IEF Cell (BioRad, Hercules, CA, USA). Isoelectric focusing was performed at 20 °C with the following conditions: increasing the voltage to 250 V for 60 min, 500 V for 60 min, 2500 V for 60 min, ramping to 4000 V for 60 min, and then focusing to reach 10,000 Vh. The strips were equilibrated for 15 min in the equilibration buffer containing 6 M urea, 2% SDS, 0.1 M Tris-HCl (pH 8), 30% glycerol, 0.5% DTT, and equilibrated for another 15 min in the equilibration buffer with the replacement of DTT with 2.5% iodoacetamide. The strips were embedded in 0.5% overlay agarose (BioRad, Hercules, CA, USA) and placed on precast acrylamide gels (Any kD Mini-PROTEAN TGX Stain-Free Protein Gels, BioRad, Hercules, CA, USA). Electrophoresis was performed at 120 V with Tris–glycine–SDS buffer (BioRad, Hercules, CA, USA). The gels were scanned with ChemiDoc MP (BioRad, Hercules, CA, USA) and protein spot analysis was performed using PDQuest Advance software, v8.0 (BioRad, Hercules, CA, USA). The quantity of each spot was normalized according to the internal standard. Protein spots that differed in expression (>2-fold) were selected and excised using ExQuest Spot-cutter (BioRad, Hercules, CA, USA) for protein identification.

### 4.9. Protein Identification by Mass Spectrometry

Protein identification experiments were performed at Kocaeli University’s DEKART Proteomics Laboratory using ABSCIEX MALDI-TOF/TOF 5800 system (Applied Biosystems, Framingham, MA, USA), as previously described [54]. Briefly, in-gel tryptic digestion for each spot was performed by using an in-gel digestion kit following the recommended protocol by the manufacturer (Pierce, Grosse Pointe, MI, USA). The digested peptide samples were desalted with a 10 μL ZipTipC18 (Millipore, Burlington, MA, USA) and eluted with the matrix solution containing 10 mg/mL α-Cyano-4-hydroxycinnamic acid and spotted onto the MALDI target plate. The TOF spectra were recorded in the positive ion reflector mode with a mass range of 400 to 2000 Da. Each spectrum was the cumulative average of 2000 laser shots. The spectra were calibrated with the trypsin auto digestion ion peaks *m*/*z* (842.510 and 2211.1046) as internal standards. Ten of the strongest peaks of the TOF spectra per sample were chosen for MS/MS analysis. The data obtained from MALDI-TOF/TOF were searched against MASCOT version 2.5 (Matrix Science) using a streamline software, ProteinPilot, v5.0.2 (ABSCIEX, Framingham, MA, USA), with the following criteria: National Center for Biotechnology Information nonredundant (NCBInr); species restriction to H. sapiens; enzyme of trypsin; at least five independent peptides matched; at most one missed cleavage site; MS tolerance set to ±50 ppm and MS/MS tolerance set to ±0.4 Da; fixed modification being carbamidomethyl (Cys) and variable modification being oxidation (Met); peptide charge of 1+ and being monoisotopic. Only significant hits, as defined by the MASCOT probability analysis (*p* < 0.05), were accepted.

### 4.10. Validation Experiments by Western Blot

Thirty micrograms of protein from each sample were diluted with 4× Laemmli buffer (BioRad, Hercules, CA, USA) and reduced with 2-mercaptoethanol (1:10). The samples were boiled at 95 °C for 5 min and loaded onto 4–15% TGX stain-free gels (BioRad, Hercules, CA, USA). Electrophoresis was performed at 60 V and gels were transferred onto PVDF membranes (BioRad, Hercules, CA, USA) in the Trans-Blot Turbo Blotting system (BioRad, Hercules, CA, USA) by applying 1 A up to 25 V for 30 min. PVDF membranes were incubated in a blocking buffer containing 20 mM Tris, 150 mm NaCl, 0.1% Tween-20, and 5% bovine serum albumin (Sigma-Aldrich, Burlington, MA, USA) for 1 h. Then, PVDF membranes were incubated at 4 °C overnight with a primary antibody for Profilin-1 (Cell Signaling, 1:2000), in addition to GAPDH (Abcam, 1:5000, Cambridge, UK) as a housekeeping protein for normalization in each assay. PVDF membranes were washed with TBS-T solution (20 mm Tris, 150 mm NaCl, and 0.1% Tween-20) and incubated for 1 h in a secondary antibody solution containing anti-rabbit (Abcam, 1:5000, Cambridge, UK) antibodies. PVDF membranes were washed again with TBS-T solution and incubated in a blotting substrate (ECL Western Blotting Substrate, Pierce, Grosse Pointe, MI, USA) according to the manufacturer’s recommendations. Protein bands were scanned in the chemiluminescent channel in ChemiDoc MP (BioRad, Hercules, CA, USA). The protein intensities were calculated by Image Lab software, v6.0.1 (BioRad, Hercules, CA, USA) and normalized according to actin intensity.

### 4.11. Statistical and Bioinformatics Analysis

Results were recorded as mean ± SD unless otherwise stated. All data except spot analysis were analyzed by Mann–Whitney test for two group comparison and Kruskal–Wallis test followed by Dunn’s multiple comparisons test on GraphPad Prism 7.0 (GraphPad Software Inc., San Diego, CA, USA). The significance level was defined as *p* < 0.05. Spot analysis was performed by two-sample t-tests assuming unequal variances with SPSS v.20.0 (IBM Corporation, Armonk, NY, USA). The significance level was defined as *p* < 0.05. Differentially expressed proteins were analyzed by PANTHER classification system (http://www.pantherdb.org) (accessed on 31 October 2023). The protein–protein interaction network of the identified proteins was constructed with the online analysis tool STRING v10.0 (http://www.string-db.org) (accessed on 31 October 2023). STRING parameters were set according to the following criteria: the type of interaction is dependent on the evidence with minimum required score of 0.4; the first shell is <10 interactors and the second shell is <5 interactors. The results were searched against Uniprot database (https://www.uniprot.org) (accessed on 31 October 2023).

## 5. Conclusions

Profilin-1, an actin binding protein, may be responsible for impaired hemostatic functions of platelets in PEX and RVO patients that lead to aberrant reorganization of the cytoskeleton during platelet activation. This novel finding could shed light on the hypercoagulable nature of the disease, although it is unclear whether reduced levels of Profilin-1 expression can lead to a prothrombotic phenotype. Hypercoagulability is also revealed by shortened clot formation time (CFT) in PEX cases, which would be associated with the risk of thrombotic events in this syndrome. Elevated soluble P-selectin levels in the plasma of PEX cases may be a more reliable marker than bound P-selectin on the platelet surface, with the lack of association with E-selectin and von Willebrand factor, indicating a major contribution of platelets to the etiopathogenesis of PEX syndrome—even more so than endothelial factors. Monitoring CFT by ROTEM and soluble P-selectin by ELISA may offer a novel assessment tool for the management of PEX patients to predict the risk for thrombotic events. The exact roles of Profilin-1 on hypercoagulable profiles of patients are worthy of future investigation to show its contribution to thrombotic events.

## Figures and Tables

**Figure 1 ijms-25-01403-f001:**
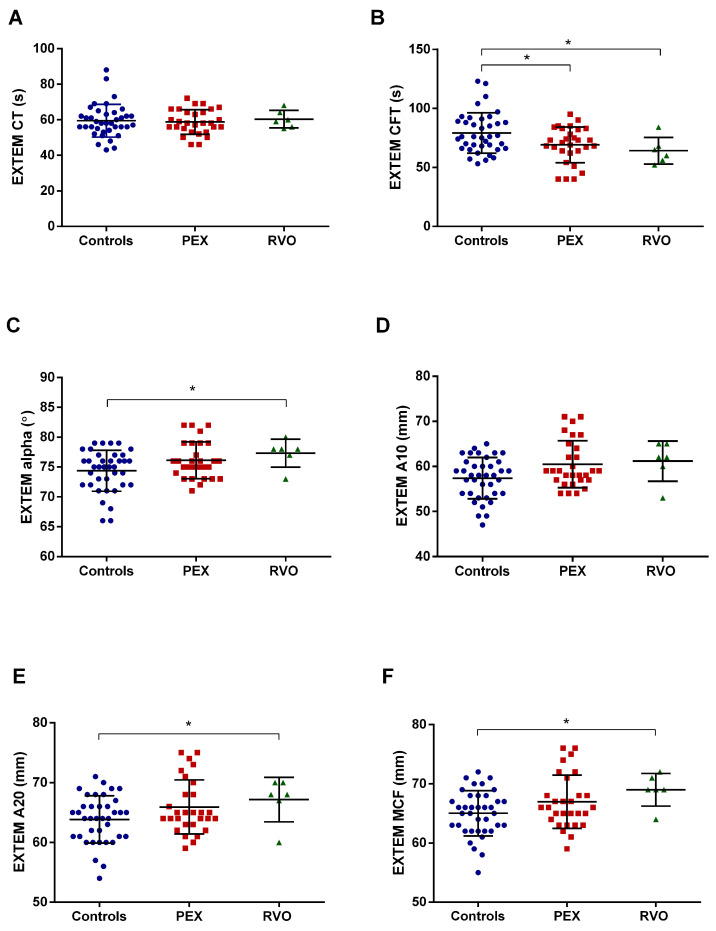
Parameters of Extem test in PEX and RVO patients, and the controls. Each parameter is depicted in a different panel. (**A**) Clotting time (CT). (**B**) Clot formation time (CFT). (**C**) Alpha angle. (**D**) A10 (amplitude at the 10th minute). (**E**) A20 (amplitude at the 20th minute). (**F**) Maximum clot firmness (MCF). * *p* < 0.05.

**Figure 2 ijms-25-01403-f002:**
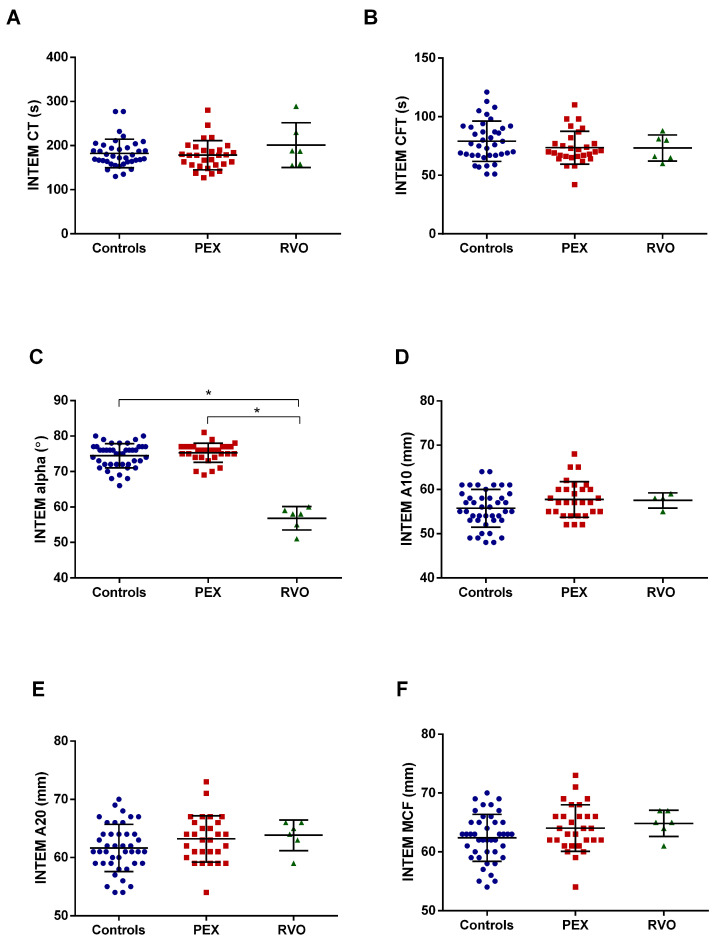
Parameters of Intem test in PEX and RVO patients, and the controls. Each parameter is depicted in a different panel. (**A**) Clotting time (CT). (**B**) Clot formation time (CFT). (**C**) Alpha angle. (**D**) A10 (amplitude at the 10th minute). (**E**) A20 (amplitude at the 20th minute). (**F**) Maximum clot firmness (MCF). * *p* < 0.05.

**Figure 3 ijms-25-01403-f003:**
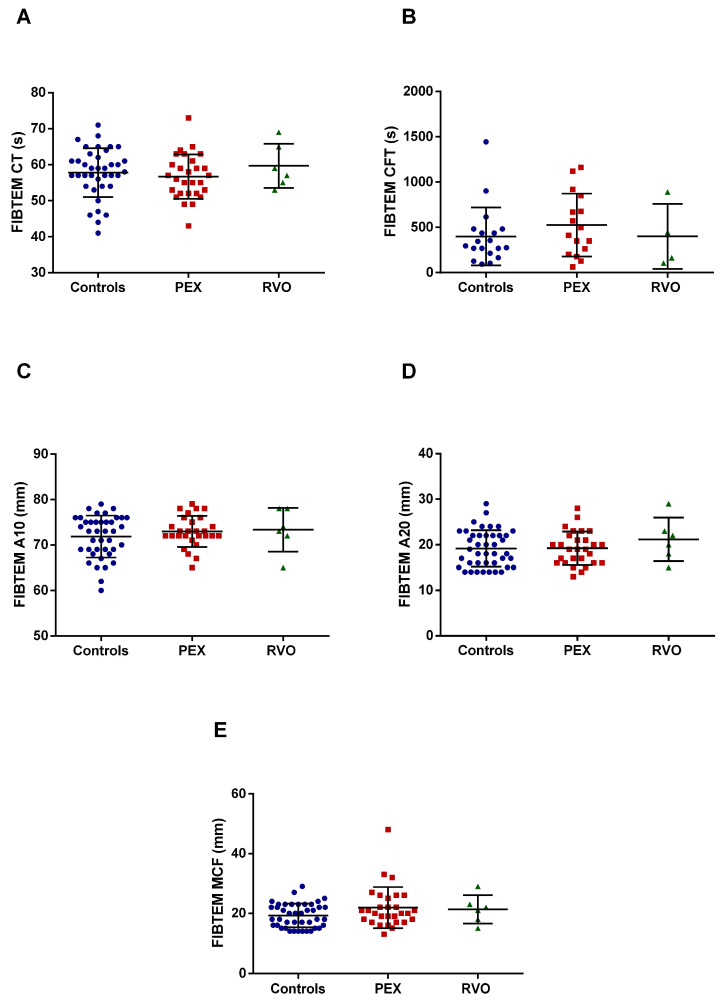
Parameters of Fibtem test in PEX and RVO patients, and the controls. Each parameter is depicted in a different panel. (**A**) Clotting time (CT). (**B**) Clot formation time (CFT). (**C**) A10 (amplitude at the 10th minute). (**D**) A20 (amplitude at the 20th minute). (**E**) Maximum clot firmness (MCF).

**Figure 4 ijms-25-01403-f004:**
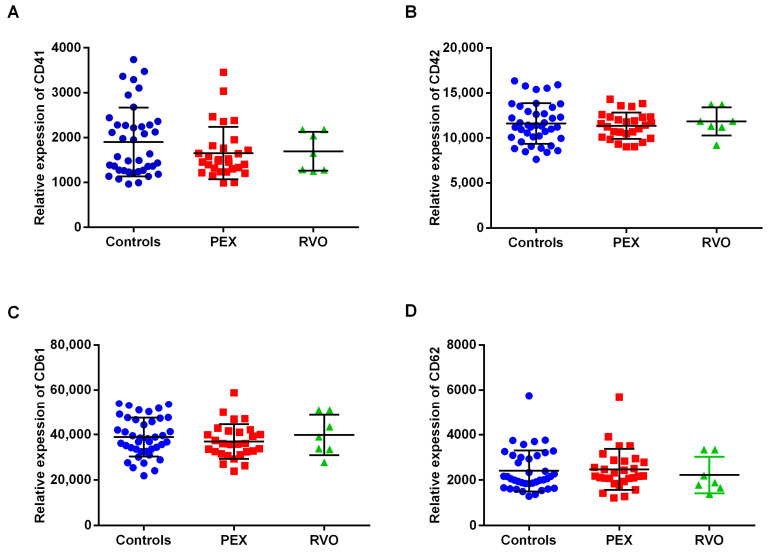
The surface expression of platelet markers in control group, pseudoexfoliation (PEX) patients, and retinal vein occlusion (RVO) patients. The relative expression levels of CD41a (**A**), CD42b (**B**), CD61 (**C**), and CD62p (**D**) markers are shown.

**Figure 5 ijms-25-01403-f005:**
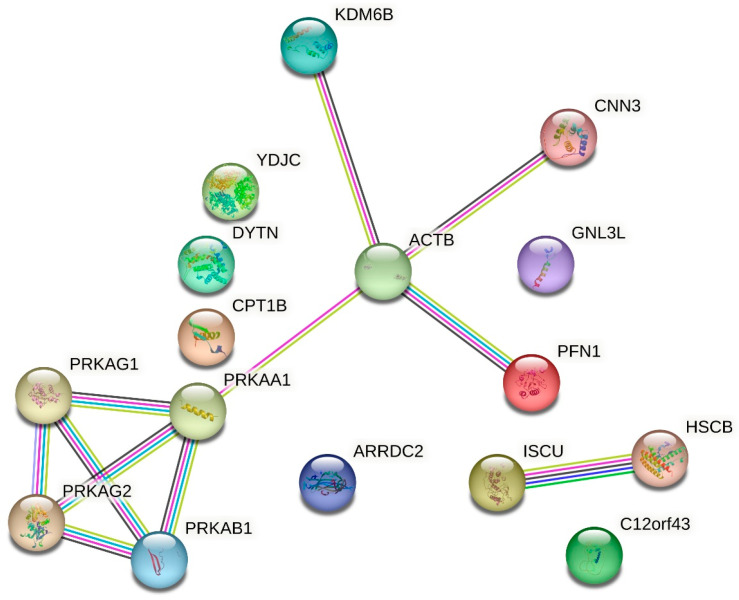
Bioinformatic analysis of protein–protein interactions of differentially expressed proteins in platelets of PEX and RVO patients, and the controls. The abbreviations of protein names are provided in Table 1, except ACTB and HSCB, which refer to beta-actin and iron–sulfur cluster co-chaperone protein, respectively. Pink and light-blue lines confirm known interactions, green and dark-blue lines indicate predicted interactions, and black lines indicate the co-expression between proteins http://www.string-db.org (accessed on 31 October 2023).

**Figure 6 ijms-25-01403-f006:**
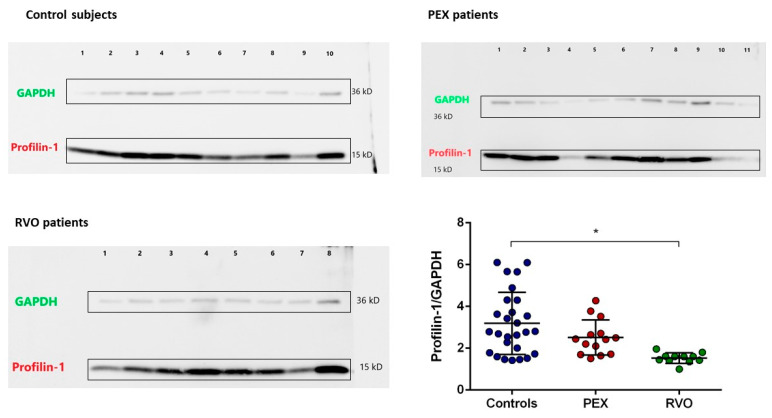
Profilin-1 expression levels in PEX and RVO patients, and the controls. Images show Profilin-1 and GAPDH expression in a representative group of patients and controls. Profilin-1 expression levels were normalized to GAPDH. The graph shows the change in relative protein expression in all studied subjects. * *p* < 0.001.

**Figure 7 ijms-25-01403-f007:**
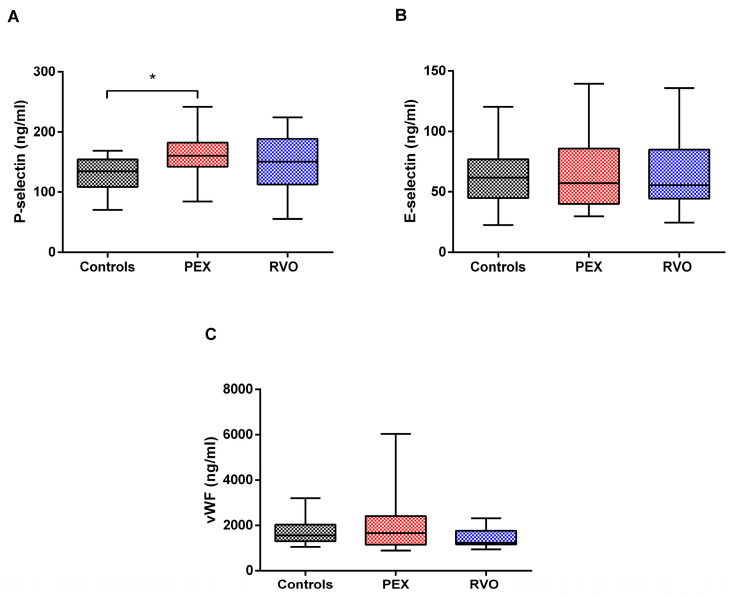
Plasma levels of endothelial activation markers in PEX and RVO patients, and the controls. (**A**) P-selectin levels, (**B**) E-selectin levels, and (**C**) von Willebrand factor (vWF) levels in study groups. * *p* < 0.05.

**Table 1 ijms-25-01403-t001:** The list of differentially expressed proteins with their respective names, UniProt and accession numbers, mass, and Mascot score values by mass spectrometry analysis.

UniProt No	Accession	Mass	Mascot Score	Protein
P07737	PROF1	15,045	38	Profilin-1
Q9H1K1	ISCU	17,926	43	Iron–sulfur cluster assembly enzyme ISCU, mitochondrial
A8MPS7	YDJC	34,444	31	Carbohydrate deacetylase
Q7Z5P9	MUC19	597,790	20	Mucin-19
O15054	KDM6B	180,299	20	Lysine-specific demethylase 6B
Q96C57	CL043	28,153	38	Protein CUSTOS (C12orf43)
Q92523	CPT1B	87,744	26	Carnitine O-palmitoyltransferase 1, muscle isoform
Q9NVN8	GNL3L	65,532	28	Guanine nucleotide-binding protein-like 3-like protein
Q96N23	CFA54	92,263	27	Cilia- and flagella-associated protein 54
A2CJ06	DYTN	65,279	22	Dystrotelin
Q8TBH0	ARRD2	44,351	21	Arrestin domain-containing protein 2
Q15417	CNN3	36,391	32	Calponin-3
Q9Y478	AAKB1 (PRKAB1)	30,363	32	5’-AMP-activated protein kinase subunit beta-1

**Table 2 ijms-25-01403-t002:** Characteristics and demographics of all study participants.

	Controls *n* = 42	PEX *n* = 29	RVO *n* = 11
Gender (f/m)	18/24	14/15	6/5
Age (year)	63 ± 7	70 ± 4	58 ± 2
Platelet count (×10^3^/µL)	257.58 ± 53.39	268.40 ± 80.8	259.71 ± 36
Mean platelet volume (MPV) (fL)	10.37 ± 0.96	10.18 ± 0.96	10.36 ± 0.81
Neutrophil/LymphocyteRatio (NLR)	2.38 ± 0.76	2.60 ± 1.04	2.51 ± 0.93
Visual acuity (log MAR)	20/22 ± 2	20/22 ± 2	20/120 ± 5 *
Intraocular pressure (mmHg)	14.95 ± 1.81	16.24 ± 2.0	14.35 ± 3.2

Data are presented as mean ± SD. * *p* < 0.001.

## Data Availability

The data presented in this study are available on request from the corresponding author. The data are not publicly available due to the confidentiality agreement with the funding institution[insert reason here].

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
