# Peer review of "Platelet Proteome Reveals Novel Targets for Hypercoagulation in Pseudoexfoliation Syndrome"

_ijms, 2024, doi:10.3390/ijms25031403_

Round 1

Reviewer 1 Report

Comments and Suggestions for Authors

In this manuscript, Ugurel et al investigated global coagulation (via ROTEM), proteome and functions of platelets in PEX patients, to determine prognostic biomarkers for thrombosis risk in PEX. The authors found, in ROTEM Extem test, significantly shortened clot formation time and clot firmness in PEX and RVO patients than healthy controls, suggesting a whole blood hypercoagulable state in PEX. Interestingly, in ROTEM FIBTEM where the platelet contribution to clot firmness is blocked, the clot firmness was similar between PEX patients and controls, indicating PEX platelets may play key roles in the increased clot firmness. Proteome analyses identified decreased expression of Profilin-1 and an increased expression of Calponin-3 in platelets from PEX patients, suggesting altered cytoskeleton function. Lastly, the authors found increased plasma p-selectin level and unchanged e-selectin or vwf level.

Comments:

1.     Recommend adding some words in the introduction section describing the role of Profilin-1 and Calponin-3 in platelet function.

2.     The baseline characteristics of studied subjects should be briefly described in the beginning of the result section. Statistic analyses results should be shown within table 3 (e.g. NLR comparisons between groups). Some of the parameters in table 3 were not presented with mean+/- SD or Median (IQR), the authors are recommended to revise that.

3.     Could the authors show representative ROTEM traces in figures 1-3? This may help readers comprehend the clotting differences between patients and controls.

4.     Does sP-selectin level correlate with ROTEM parameters?

5.     Could the authors briefly comment on why no correlation was found between platelet Profilin-1 and Calponin-3 levels with ROTEM parameter?

Reviewer 2 Report

Comments and Suggestions for Authors

The authors describe the differences observed in coagulation and platelet proteome, in patients with pseudoexfoliation and retinal vein occlusion, compared to controls. They claim to have identified two potential targets for therapy to ameliorate thrombotic events in PEX patients, from the protemics data.

The manuscript is well written, in general.

However, the reviewer has some concerns and suggestions.

Abstract: 

line 20: the platelet markers are not activation markers. They are all platelet surface receptor markers, except for CD62P, which is P-selectin, and it indicates degranulation of alpha granules when translocated to the membrane. Also correct this misleading description of markers in section 2.2.

General conceptual:

The authors conclude that 2 proteins differ in PEX, but actually they do not differ from controls, so this reviewer wonders how these proteins can be considered targets for treatment to avoid thrombotic events in PEX patients...

Why was 2D-gel separation used? The reviewer would advise to run MS on samples that have been processed in solution directly (without running samples in 1D or 2D gels). Furthermore, why pooling samples? it would be better to run the independent samples, and then apply statistics for significance.

The authors claim that the differences in the proteins mentioned causes conformational changes in platelets, and hence their prothrombotic phenotype. However, this is speculation and has not ben shown at all.

Round 2

Reviewer 1 Report

Comments and Suggestions for Authors

Thanks for revising the paper. i have no more comments.

Reviewer 2 Report

Comments and Suggestions for Authors

I think the author´s have answered the raised issues. The manuscript has improved.